



# How to reconstruct diffuse radiation scenario for simulating GPP in land surface models?

Yuan Zhang[1,2], Olivier Boucher[3], Philippe Ciais[1], Laurent Li[2], and Nicolas Bellouin[4]

[1]Laboratoire des Sciences du Climat et de l'Environnement (LSCE), IPSL, CEA/CNRS/UVSQ, Gif sur Yvette, France
[2]Laboratoire de Météorologie Dynamique, IPSL, Sorbonne Université/CNRS, Paris, France
[3]Institut Pierre–Simon Laplace, CNRS/Sorbonne Université, Paris, France
[4]Department of Meteorology, University of Reading, Reading, RG6 6BB, UK

**Correspondence:** Yuan Zhang (yuan.zhang@lmd.jussieu.fr)

**Abstract.** The impact of diffuse radiation on photosynthesis has been widely documented in field measurements. This impact may have evolved over time during the last century due to changes in cloudiness, increased anthropogenic aerosol loads over polluted regions, and to sporadic volcanic eruptions curtaining the stratosphere with sulfate aerosols. The effect of those changes in diffuse light on large-scale photosynthesis (GPP) are difficult to quantify, and land surface models have been de-

signed to simulate them. Investigating how anthropogenic aerosols have impacted GPP through diffuse light in those models requires carefully designed factorial simulations and a reconstruction of background diffuse light levels during the pre-industrial period. Currently, it remains poorly understood how diffuse radiation reconstruction methods can affect GPP estimation and what fraction of GPP changes can be attributed to aerosols. In this study, we investigate different methods to reconstruct spatio-temporal distribution of the fraction of diffuse radiation (Fdf) under pre-industrial aerosol emission conditions using a

land surface model with a two-stream canopy light transmission scheme that resolves diffuse light effects on photosynthesis in a multi-layered canopy, ORCHIDEE_DF. We show that using a climatologically-averaged monthly Fdf, as has been done by earlier studies, can bias the global GPP by up to 13 PgC yr$^{-1}$ because this reconstruction method dampens the variability of Fdf and produces Fdf that is inconsistent with short-wave incoming surface radiation. In order to correctly simulate pre-industrial GPP modulated by diffuse light, we thus recommend that the Fdf forcing field should be calculated consistently with synoptic,

monthly and inter-annual aerosol and cloud variability for pre-industrial years. In the absence of aerosol and cloud data, alternative reconstructions need to retain the full variability in Fdf. Our results highlight the importance of keeping consistent Fdf and radiation for land surface models in future experimental designs that seek to investigate the impacts of diffuse radiation on GPP and other carbon fluxes.





## 1 Introduction


Gross primary production (GPP) is one of the largest carbon fluxes in the global carbon cycle and the only way by which land ecosystems capture $CO_2$ from the atmosphere. The GPP of terrestrial ecosystems is known to be sensitive to climate factors including temperature, precipitation and incoming shortwave radiation (Nemani *et al.*, 2003). During the recent two decades, several *in situ* studies reported that in addition to the total amount of incoming shortwave radiation, the fraction of diffuse

radiation (Fdf) as a part of the total radiation can also strongly affect GPP and in turn the carbon cycle (Gu *et al.*, 2002, 2003; Niyogi *et al.*, 2004). For a given level of incoming radiation, conditions with more diffuse light are found to increase light use efficiency by 6-180% in different vegetation types (Alton *et al.*, 2007; Choudhury, 2001; Gu *et al.*, 2002), which will increase the total GPP. This is because that the diffuse radiation can penetrate deeper and reach more shaded leaves in the deep canopy and consequently enhance the canopy photosynthesis. In other words vegetation is sensitive to both light quantity and quality.

This effect of diffuse radiation is potentially an important explanation of observed large-scale trends of GPP because the aerosol emissions from anthropogenic activities have increased, and the light-scattering properties of those aerosols augment the diffuse fraction of light. In addition, sporadic volcanic eruptions inject aerosols in the stratosphere which decrease the amount of light reaching the surface and strongly increase its diffuse fraction globally during a few years after each eruption. However, the large scale impacts of aerosol-induced light quality changes remain poorly quantified. The recent development

of land surface models (LSMs) that distinguish direct and diffuse light in canopy light transmission (Dai *et al.*, 2004; Alton *et al.*, 2007; Mercado *et al.*, 2009; Chen, 2013; Yue and Unger, 2017; Zhang *et al.*, 2020) provides an opportunity to study how diffuse light and other climate variables affect GPP and its variability.

In spite of the increasing number of LSMs considering diffuse light, there remains no standard experimental design for isolating the impacts of aerosol-induced diffuse radiation changes on GPP. Alton *et al.* (2007) compared equivalent simulations

performed with two LSMs, one with a one-stream and the other one with a two-stream canopy light transmission scheme accounting for diffuse and direct light effects on GPP. In contrast, Mercado *et al.* (2009) designed, two scenarios with and without changes in Fdf, using the same LSM, keeping all other climate forcing variables identical. Currently, the lack of harmonized design for modeling GPP from diffuse light still prevents the comparison of those different results to understand the magnitude and uncertainties of aerosol impacts. Therefore, a rigorous simulation design for LSMs resolving diffuse light

effects is urgently needed.

Because the one-stream and two-stream canopy light transmission models do not necessarily give the same GPP when there is no changes in Fdf, it is difficult to interpret the GPP difference detected by the two LSMs in Alton *et al.* (2007) as impacts of diffuse radiation changes. In contrast, considering factorial simulations with the same LSM capable of resolving diffuse light effects, underwith different diffuse light forcing scenarios, e.g., historical aerosol emissions and pre-industrial

aerosol emissions, removes the interference from different model structures. However, attention has to be paid on how to define the forcing of pre-industrial aerosols in the baseline simulation. Currently, there are mainly two possible approaches to reconstruct pre-industrial diffuse light forcing, given gridded fields of all other climate variables needed as input for a given LSM. The first approach relies on the climate and diffuse light fields from Earth system models (ESMs) runs with and without





anthropogenic aerosol emission scenarios. This approach can provide a full set of climate variables without anthropogenic

aerosols, which defines the baseline, but suffers from large climate biases and uncertainties from ESMs, which inevitably leads

to large biases in the modeled GPP (Zhang *et al.* , 2019). Furthermore, ensemble simulations with ESMs may also be required

to detect and attribute the impacts of anthropogenic aerosols from natural climate variability, which unavoidably arises when

different simulations are performed. Such a detection/attribution framework (Eyring *et al.* , 2016) is used for attributing the

effect of human induced radiative forcing on climate change, but requires large ensemble of ESM simulations, often at the cost

of reduced spatial resolution. The other approach relies on using reconstructed gridded climate fields based on observations

and adding to these fields the variability of the Fdf. Compared to using ESM climate, the reconstructed climate is more

accurate. However, a counterfactual reconstruction with constant pre-industrial aerosols during the entire historical period is

not available. To investigate the anthropogenic aerosol impacts, such a "pre-industrial or no-anthropogenic aerosol scenario"

must thus be reconstructed based on careful assumptions.

For the sake of isolating the impacts of aerosol-induced light quality changes, the problem is thus to reconstruct a no-

anthropogenic-aerosol multi-year baseline pre-industrial forcing that keeps the Fdf at the pre-industrial level but retains the

natural variation of shortwave light and of all other climate fields. Mercado *et al.* (2009) opted to prescribe a monthly mean

climatology of Fdf in their pre-industrial baseline scenario. This is a coarse approximation because Fdf has a strong covariance

with all other climate variables, especially shortwave radiation. For instance, a sunny month of January in a given year cannot

have the same mean Fdf value as a very cloudy January in another year. Similarly, a sunny 1$^{st}$ of July in one grid-cell cannot be

assigned the same Fdf as a cloudy 1$^{st}$ of July happening another year. The averaging used by Mercado *et al.* (2009) inevitably

causes a mismatch between Fdf and other climate forcing variables. Considering the non-linear response of GPP to Fdf for

different climate forcing conditions, the monthly mean climatology approach to reconstruct pre-industrial Fdf may cause biases

in the baseline GPP and consequently on the attribution of the historical anthropogenic aerosol impacts on GPP changes exam-

ined against this baseline. In this study, we study a set of simulations using different diffuse light reconstructions to evaluate

the impacts of the reconstruction method on the simulated anthropogenic-aerosol impacts on GPP during the historical period

(1901–2012) using the recently developed ORCHIDEE-DF LSM which has a two-stream canopy light transmission scheme

and accounts for Fdf and climate effects on GPP over the whole globe. The main objectives of this study are the following:

i) investigate whether and by how much the Fdf baseline pre-industrial reconstruction method affects GPP; ii) identify the

underlying mechanisms of the modeled GPP dependence upon the chosen reconstruction method; and iii) recommend a best

reconstruction method for future studies, that could be adopted by other LSMs resolving diffuse light impacts.

## 2 Data and Methods

### 2.1 Model Description

In order to simulate the impact of diffuse radiation, we used the ORCHIDEE_DF LSM, which is originally based on the

ORCHIDEE LSM trunk (v5453) (Krinner *et al.* , 2005), but has a two-stream canopy radiation transmission module to dis-

tinguish direct and diffuse radiation (Zhang *et al.* , 2020). ORCHIDE_DF has been evaluated using obervations from over





159 flux sites over the globe and was proven to be able to reproduce the observed GPP sensitivity to diffuse light (Zhang *et al.* , 2020) under the same light and other climate fields conditions. Instead of using the empirical light partitioning module of the original ORCHIDEE_DF that calculated Fdf from shortwave radiation and solar angle (Zhang *et al.* , 2020), we mod-

ified the model to let it read Fdf from gridded forcing files, along with other climate variables. Due to the new canopy light transmission scheme, the model need to be recalibrated to obtain C fluxes that match observation-based estimations. Here, we empirically tuned the photosynthesis-related parameters (Vcmax, specific leaf area, leaf age) within some reasonable ranges to simulate similar GPP as in the TRENDY V8 S3 simulation performed with the ORCHIDEE trunk version for each plant functional type and during the 1900s. We chose the TRENDY V8 S3 simulation as the reference because the ORCHIDEE trunk

version for this simulation has been well-tuned to simulate C fluxes matching the observation-based global carbon budget (http://sites.exeter.ac.uk/trendy/), also due to the easy accessibility of data.

## 2.2   Forcing data and experimental design

The climate forcing used in this study is the 6-hour CRUJRA v1.1 dataset (Harris *et al.* , 2014; Harris, 2019; Kobayashi, 2015). CRUJRA v1.1 dataset was generated by adjusting the Japanese Reanalysis data (JRA) produced by the Japanese Meteorological

Agency (JMA) with the observationally-based monthly Climatic Research Unit (CRU) TS 3.26 data. It provides 6-hourly meteorological variables at $0.5 \times 0.5°$ resolution including 2-metre air temperature, total precipitation, downward shortwave radiation, downward longwave radiation, 2-meter specific humidity, air pressure and the zonal and meridional components of the 10-m wind. This dataset is the standard forcing input used in the TRENDY v7 simulations (http://sites.exeter.ac.uk/trendy/).

For the sake of investigating the effect of diffuse radiation with a framework consistent with the TRENDY simulation

protocol (http://sites.exeter.ac.uk/trendy/), a new Fdf field was calculated along with the above-mentioned climate variables at the same spatial and temporal resolutions. The Fdf field is based on atmospheric radiative transfer calculations using global aerosol and cloud cover fields, including tropospheric aerosol optical depth for the period 1900-2017 from the HadGEM2-ES CMIP5 historical simulation (Bellouin *et al.* , 2011) bias-corrected to match the averaged speciated aerosol optical depths simulated over the period 2003-2017 by the CAMS reanalysis of atmospheric composition (Inness *et al.* , 2019), the updated

climatology of stratospheric aerosol optical depth by Sato *et al.*  (1993), extended from 1850-2012 to 2017 by replicating the quiescent year 2010 to represent 2013-2017 and the JRA cloud fraction (Kobayashi, 2015) itself bias-corrected by the observation-based CRU TS v4.03 monthly cloud data.

To evaluate the methods reconstructing baseline Fdf under pre-industrial aerosol conditions, we first selected only the volcano-free years during 1901-1920 (Table 1) when there were negligible volcanic aerosol emissions and anthropogenic

aerosol emissions were about a third of their present-day rates to affect Fdf. Based on the assumption that this sample is representitive to the pre-industrial aerosol conditions, four methods are used to reconstruct 0.5° x 0.5° 6-hourly pre-industrial Fdf field. The first method, noted as *DF-PI-MON-CLIM* , is based on a monthly climatology mean, i.e. all the 6-hour time steps within a certain month take the same value from the mean Fdf of this month across the selected years. This method is similar to the approach used by Mercado *et al.*  (2009). The second method accounts for the fact that there is a strong diurnal cycle

of Fdf. This diurnal cycle is important because the diffuse light impact on GPP is not the same at different time of a day due





to different radiation levels. In order to retain the diurnal cycle of Fdf, the second method, named *DF-PI-6H-CLIM* , uses a 6-hour climatological mean across the selected years:

$$Fdf_f(t) = \frac{\sum_y Fdf_o(y,t)}{\sum_y y} \qquad (1)$$

where $Fdf_f(t)$ is the final Fdf at time step $t$, $Fdf_o(y,t)$ is the original Fdf at time step $t$ of year $y$. This method accounts for
the periodical diurnal increase of Fdf from morning to mid-day and its decrease from mid-day to afternoon, but it ignores the variability of Fdf between years. For instance, the same time step may be very sunny with low Fdf in one year but completely cloudy with high Fdf in other. The *DF-PI-6H-CLIM* reconstruction smooths the Fdf and give medium Fdf for both years, which is not realistic (Fig. 1). The third method avoids this smoothing of the variability of Fdf and uses Fdf directly from years select from 1901-1920 to have similar average annual Fdf as the previous two reconstructions. Because the Fdf variation differs
among years, to understand the uncertainty caused by this difference, we use an ensemble of three members, *DF-PI-1901*, *DF-PI-1905* and *DF-PI-1916*, which respectively repeat the Fdf field of 1901, 1905 and 1916 to the entire simulation period. The final estimation of C fluxes are based on the average of the output of the three members. This reconstruction based on ensemble simulations is named *DF-PI-ENS* . Finally, a new Fdf field, *DF-PI-AERO* , is generated using the same atmospheric radiative transfer model previously described but the anthropogenic emissions were kept at the 1901-1920 level and the volcanic aerosols
emissions were excluded.

Besides the above-mentioned reconstructions, a historical simulation (*DF-HIST* ) driven by the original Fdf is set up as the reference to investigate the impacts of diffuse radiation. Except the Fdf field, all these simulations use the same climate and land use maps which vary throughout the simulations. Also, all these simulations start from the same state of a spin-up simulation who has equilibrated the C pools using 1901-1920 climate and Fdf. A detailed description of each simulation can
be found in Table 1.

## 3 Results

### 3.1 Impacts of Fdf changes at the global scale

As shown in Fig. 2a, the historical global mean Fdf has three phases during the entire study period. Before 1950, the mean Fdf varies around 0.615-0.62. During 1950-1980, the mean Fdf increases from 0.62 to 0.64 mainly in response to increasing
anthropogenic aerosol emissions (Lamarque *et al.* , 2014). After 1980s, the mean Fdf stays around 0.64. In addition to these three phases, notable spikes of Fdf of 0.02-0.04 are found in years with strong volcanic eruptions, the Santa Maria in 1902-1903, El Chichón in 1982, Mount Pinatubo in 1991. Because all the no-anthropogenic-aerosol reconstructions use the volcano-free years (*DF-PI-6H-CLIM* , *DF-PI-ENS* , *DF-PI-MON-CLIM* ) or do not include volcanic aerosols (*DF-PI-AERO* ) the during 1901-1920, they produce the same or very similar global yearly mean Fdf around 0.615 during the entire study period with
inter-method differences smaller than 0.002 (not shown), which is much smaller than the variability in historical Fdf.

In response to the different interannual variation of Fdf between the *DF-HIST* and no-anthropogenic-aerosol scenarios, the differences between *DF-HIST* and all no-anthropogenic-aerosol GPP ($\Delta$GPP) also show a 3-phase pattern (Fig. 2b), with the





**Table 1.** Experimental design in this study

| | Name | Climate and land use maps | Incoming diffuse shortwave radiation fraction |
|---|---|---|---|
| | *spinup*[1] | 1901-1920 cycling | 1901-1920 cycling |
| | *DF-HIST* | All variables varying | Varying during the study period |
| | *DF-PI-6H-CLIM* | All variables varying | Repeat the 6 hour average of 1901, 1904-1906, 1909, 1911, 1915-1920, with diurnal and seasonal variations maintained |
| | *DF-PI-MON-CLIM* | All variables varying | Repeat the monthly average of 1901, 1904-1906, 1909, 1911, 1915-1920 over all years |
| *DF-PI-ENS*[2] | *DF-PI-1901* | All variables varying | Repeat the field of 1901 |
| | *DF-PI-1905* | All variables varying | Repeat the field of 1905 |
| | *DF-PI-1916* | All variables varying | Repeat the field of 1916 |
| | *DF-PI-AERO* | All variables varying | Calculated using atmospheric light transfer model with 1901-1920 aerosol level (volcanic aerosols excluded) |

[1] All the other simulations start from the stage when C pools are equilibrated (340-yr simulation using the spinup-analytic simulation in ORCHIDEE_DF)

[2] The average of *DF-PI-1901*, *DF-PI-1905*, *DF-PI-1916*

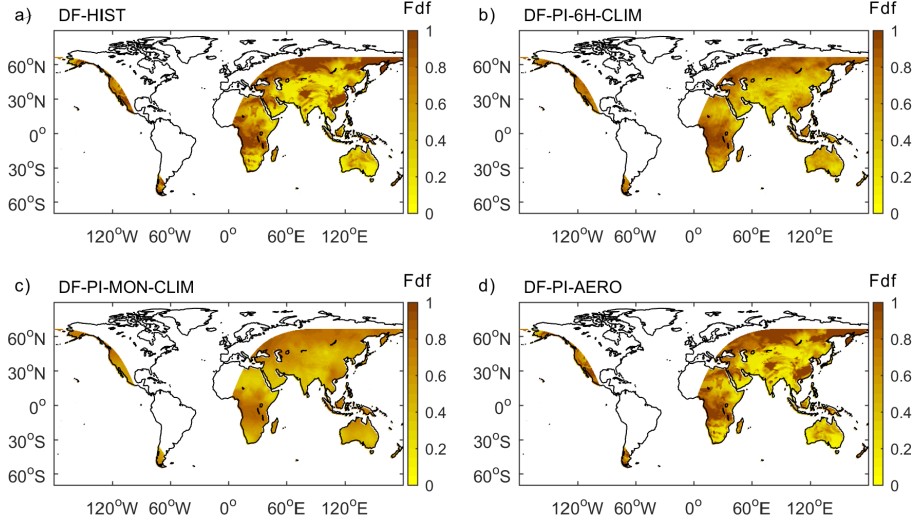

**Figure 1.** Fdf during 1901-01-01 00:00-06:00 UTC from (a) the original forcing including climate variability from CRU-JRA and historical aerosol concentration (*DF-HIST* ), (b) the 6-hour mean of 01-01 00:00-06:00 Fdf over 1901-1920 non-volcanic years (*DF-PI-6H-CLIM* ), (c) the monthly mean of January Fdf over 1901-1920 non-volcanic years (*DF-PI-MON-CLIM* ), (d) the reconstruction using climate variability from CRU-JRA and mean 1901-1920 aerosol concentration (*DF-PI-AERO* ). Night time pixels are masked for comparison.





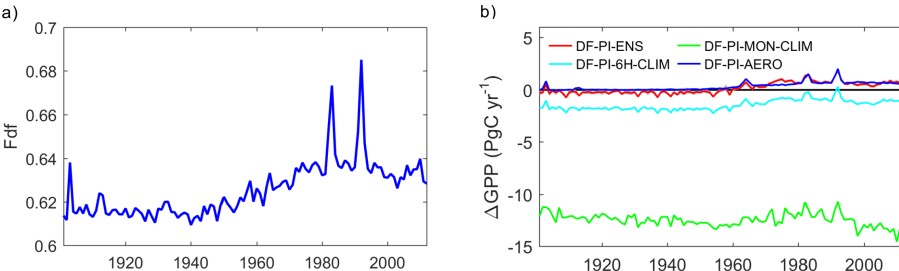

**Figure 2.** Time series of (a) global mean Fdf and (b) ΔGPP between *DF-HIST* and no-anthropogenic-aerosol scenarios. The shaded area along the red curve in (b) indicates the range of the three ensemble members of the *DF-PI-ENS* simulations.

highest ΔGPP occurring after 1960s and during large volcanic eruptions. Although the interannual variation of ΔGPP is similar among reconstructions, large discrepancies on the magnitude of global ΔGPP are found. The *DF-PI-ENS* and *DF-PI-AERO*

reconstructions show similar global GPP compared with the *DF-HIST* scenario before 1950s. In contrast, the *DF-PI-6H-CLIM* reconstruction leads to a negative ΔGPP of –1.8 PgC yr⁻¹ during the same period (1901-1950). The *DF-PI-MON-CLIM* reconstruction results in a much larger negative ΔGPP of over –12 PgC yr⁻¹. Because during 1901-1920, the Fdf of *DF-HIST* and no-anthropogenic-aerosol scenarios are at similar level, the negative ΔGPP from *DF-PI-6H-CLIM* and *DF-PI-MON-CLIM* must be related to the difference from the method chosen for the pre-industrial Fdf reconstruction.

## 3.2    spatial distribution of ΔGPP

Fig 3 shows the spatial patterns of ΔGPP derived from each reconstruction during the period (1961-2020) when Fdf is most different from the pre-industrial level. Among the four reconstructions, *DF-PI-AERO* and *DF-PI-ENS* reconstructions show positive ΔGPP of over 10 gC m⁻² yr⁻¹ in East and South Asia, Europe and tropical rainforest regions. In spite of this similarity in pattern, *DF-PI-ENS* reconstruction shows higher ΔGPP than PIaero in the West Amazon and Congo basins. Besides, the

*DF-PI-ENS* reconstruction has negative ΔGPP in high latitudes and in small patches in eastern Brazil and Uruguay, while the *DF-PI-AERO* shows much smaller regions with negative ΔGPP . In contrast to the positive ΔGPP of *DF-PI-AERO* and *DF-PI-ENS* , the *DF-PI-6H-CLIM* reconstruction shows negative ΔGPP of –10 to –40 gC m⁻² yr⁻¹ in East US, West Europe, South China and large regions of South America. The *DF-PI-MON-CLIM* reconstruction has even larger negative ΔGPP , with magnitude larger than 40 gC m⁻² yr⁻¹, over almost all vegetated regions.

## 3.3    Seasonal and diurnal variations of ΔGPP

Because the different Fdf variabilities from different reconstructions can lead to different variations in ΔGPP , we compared the seasonal and diurnal variations of ΔGPP from those different reconstructions at different latitudes (Fig. 4 and 5). At the seasonal scale, the ΔGPP from *DF-PI-AERO* is neutral during 1901-1920 at all seasons while all other reconstructions found negative ΔGPP during this time (Fig. 4). During 1993-2012, the *DF-PI-AERO* and *DF-PI-ENS* reconstructions show

remarkable positive dGPP in low latitudes, while the ΔGPP of the other two reconstructions remain negative in most latitudes.





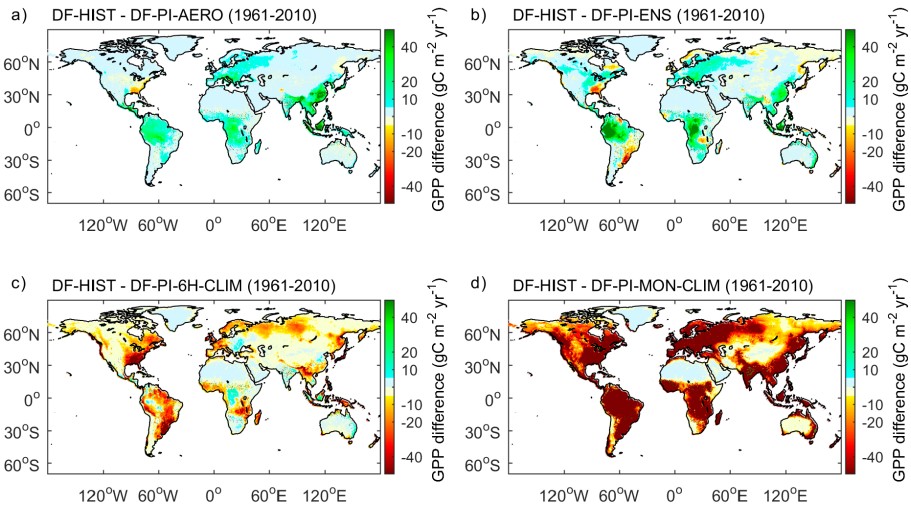

**Figure 3.** Spatial patterns of ΔGPP (in gC m⁻² yr⁻¹) between *DF-HIST* and (a) *DF-PI-AERO* , (b) *DF-PI-ENS* , (c) *DF-PI-6H-CLIM* and (d) *DF-PI-MON-CLIM* during 1961-2010.

The seasonal variations of ΔGPP from the different reconstructions are generally small compared to the latitude variability of ΔGPP derived from each experiment.

In terms of the diurnal cycle of ΔGPP , different reconstructions show very different patterns (Fig. 5). The ΔGPP from the *DF-PI-AERO* reconstruction shows remarkable positive values in the low latitudes during 2001-2005, with the largest ΔGPP

in the late morning. Similarly, the *DF-PI-ENS* simulations also show positive ΔGPP in late morning in low latitudes during the same period. However, in midlatitudes of the Southern Hemisphere, the *DF-PI-ENS* ΔGPP is negative. In contrast to *DF-PI-AERO* and *DF-PI-ENS* , the *DF-PI-6H-CLIM* reconstruction has negative ΔGPP in most latitudes and the largest ΔGPP mainly occurs in the morning. Because the *DF-PI-MON-CLIM* reconstruction smoothed out the diurnal cycle of Fdf, its ΔGPP diurnal cycles show very different pattern compared with other reconstructions. In almost all latitudes, the *DF-PI-MON-CLIM*

ΔGPP show large negative values in the afternoon. While in the morning, positive ΔGPP is found in latitudes 10°N-30°N and around 30°S.

## 4 Discussion

### 4.1 How does Fdf affect GPP?

The changes of diffuse radiation in natural conditions are often caused by changes in aerosols or cloud cover. Although many

previous studies have reported larger light use efficiency under cloudier conditions or conditions with more aerosols, it is only recently that it has been fully appreciated that this enhancement in light use efficiency is due to both changes in Fdf and other climate factors such as air temperature and VPD (Cheng *et al.* , 2015; Wang *et al.* , 2018; Zhang *et al.* , 2020) that covary




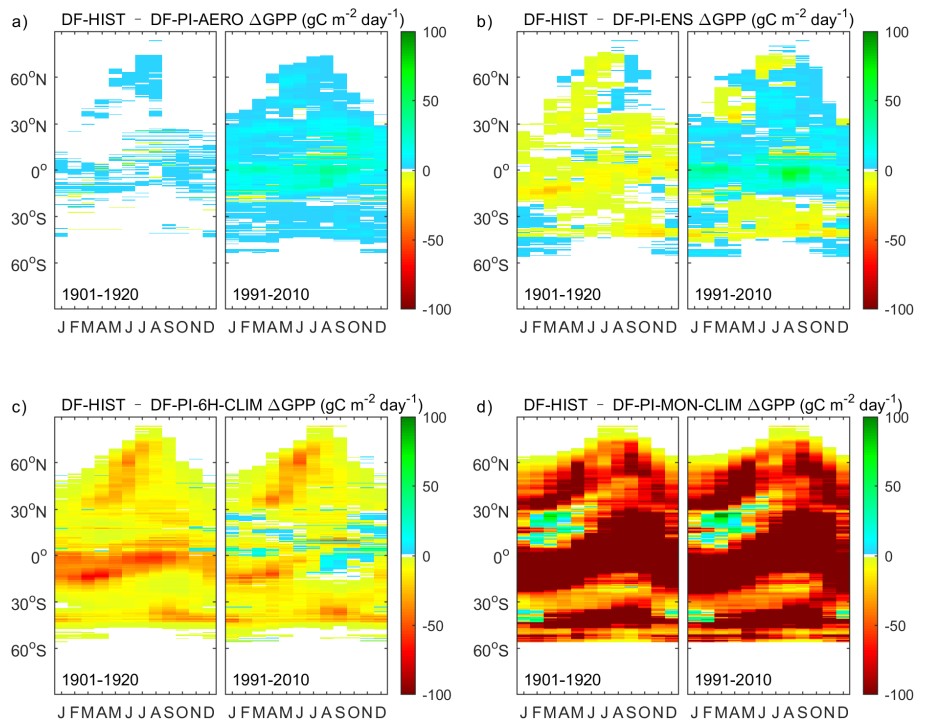

**Figure 4.** Seasonal variations of $\Delta$GPP in gC m$^{-2}$ day$^{-1}$ between *DF-HIST* and (a) *DF-PI-AERO* , (b) *DF-PI-ENS* , (c) *DF-PI-6H-CLIM* and (d) *DF-PI-MON-CLIM* during 1901-1920 and 1991-2010.

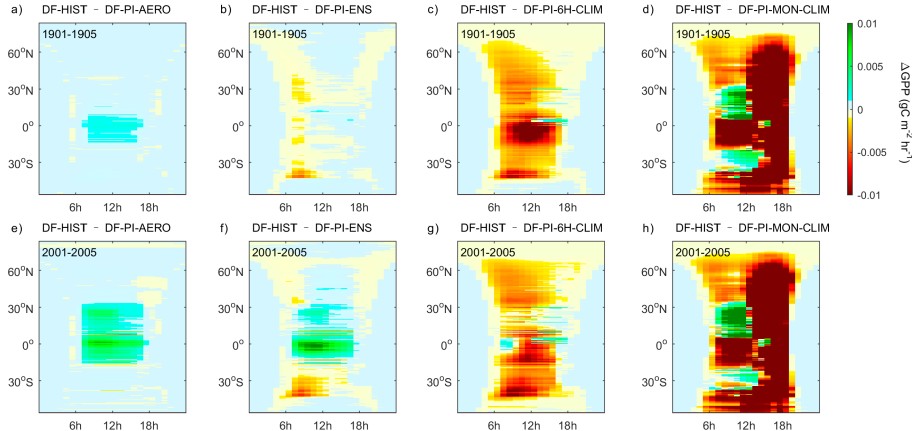

**Figure 5.** Diurnal variations of $\Delta$GPP in gC m$^{-2}$ hr$^{-1}$ between *DF-HIST* and *DF-PI-AERO* (a,e), *DF-PI-ENS* (b,f), *DF-PI-6H-CLIM* (c,g) and *DF-PI-MON-CLIM* (d,h) during 1901-1905 (a-d) and 2001-2005 (e-h).

with Fdf. In this study, we investigated the impact of Fdf alone for which there are several explanations (e.g., (Roderick *et al.* , 2001; Kivalov and Fitzjarrald, 2019)). Among these, the most widely accepted explanation is that compared with direct





radiation which can be more easily blocked by leaves in the upper part of the canopy, diffuse radiation has more chance to get absorbed by shaded leaves (Li *et al.* , 2014), especially in thick canopies with a large leaf area index. Therefore, a larger Fdf will lead to more homogeneous distribution of light in canopy. Because the light-photosynthesis response curve at leaf level is a concave function (i.e., the mean photosynthesis rate of two light levels is smaller than the photosynthesis rate at the light level equal to the average of the two). Due to this mechanism, the impacts of Fdf on GPP should be larger when there are more shaded leaves in canopy (larger LAI) and when more sunlit leaves are light-saturated (stronger incoming shortwave radiation). This generally explains the spatial pattern of ΔGPP detected in this study (Fig. 3a).

## 4.2 Why using a climatological average of the diffuse light fraction to force a LSM results in a negative bias of pre-industrial GPP?

The impacts of Fdf on GPP depend on the level of radiation, therefore, it is necessary to get consistent Fdf and radiation forcing on 6-hourly to multi-annual time scales to correctly simulate GPP and consequently the Fdf impacts. However, there is no statistical method to keep the consistency of Fdf and radiation in a counterfactual no-anthropogenic-aerosol scenario. Compared with the *DF-HIST* scenario, the *DF-PI-MON-CLIM* Fdf reconstruction averaged out the diurnal cycle of Fdf. Because the solar zenith angle is large due to longer light path in atmosphere in the morning and afternoon, the Fdf is usually large in the morning and afternoon but low at midday (Iziomon and Aro , 1998). Prescribing the same monthly average of Fdf each 6-hourly time step in the *DF-PI-MON-CLIM* reconstruction thus underestimates the Fdf in the morning and afternoon when the radiation is low and atmosphere scattering makes light predominantly diffuse, and overestimates the midday Fdf when the radiation is high. Thus, the use of the *DF-PI-MON-CLIM* method can cause a higher GPP during daytime but has a marginal impact on GPP in early morning and late afternoon. At global scale, this overestimation of GPP lead to a –12 PgC yr$^{-1}$ ΔGPP (Fig. 2), much smaller than all the other reconstructions. It should be noted that the original 6-hourly Fdf data does not cover all timesteps. The model filled the absent value of Fdf with a unity (1) value (i.e. all radiation is diffuse) when the sun is below the horizon and interpolated this value to 30-min time steps for GPP calculation. In the *DF-PI-MON-CLIM* Fdf reconstruction, all time steps are filled with an average value. If the absent values happen to be before dawn, the data-filling procedure may result in spurious positive ΔGPP (Fig. 5d,h). This artifact is expected to get corrected when the Fdf field is provided at higher temporal resolution or if better interpolation techniques are used. Nevertheless, this regional positive ΔGPP does not alter the global negative ΔGPP detected by the *DF-PI-MON-CLIM* reconstruction (Fig. 2).

Compared with *DF-PI-MON-CLIM* , the *DF-PI-6H-CLIM* reconstruction did not smooth the diurnal cycle of Fdf. However, the GPP under *DF-PI-6H-CLIM* reconstruction is still overestimated (Fig. 2). This overestimation is also from the smooth of Fdf, not the diurnal cycle but the interannual variability of Fdf. Because the Fdf is affected by the scattering of aerosols and clouds, for a given solar zenith angle, the Fdf should be negatively correlated to the total radiation reaching the canopy (Spitters, 1986; Weiss and Norman, 1985). Due to this negative relationship between radiation and Fdf, the average of Fdf at the same time over years (solar position constant) actually underestimates the Fdf under most low radiation conditions but overestimates the Fdf under most high radiation conditions. As shown in Fig. 1, there are much few cases of extremely sunny (Fdf<0.3) or extremely cloudy (Fdf>0.9) conditions in the *DF-PI-6H-CLIM* reconstruction (Fig. 1b) than in the original Fdf

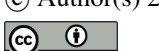


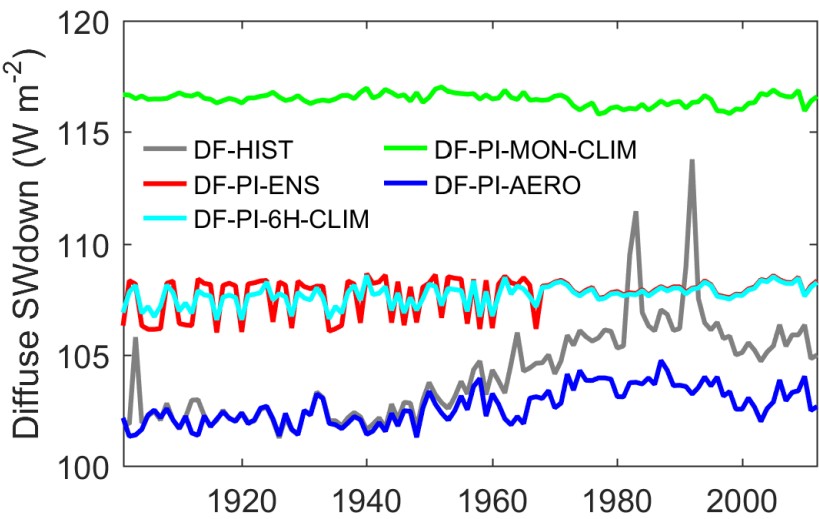

**Figure 6.** Mean diffuse incoming shortwave radiation at the surface from each reconstruction.

field (Fig. 1a). As a result, the smoothing of the Fdf interannual variability in the *DF-PI-6H-CLIM* reconstruction causes an
overestimation of the total GPP in a similar way as for the *DF-PI-MON-CLIM* reconstruction.

In contrast to the *DF-PI-6H-CLIM* and *DF-PI-MON-CLIM* reconstructions, the *DF-PI-ENS* simulations does not smooth
Fdf. As a result, the Fdf mismatch with radiation is independent from the radiation level, although the Fdf remains inconsistent
with the synoptic and inter-annual variability of shortwave light and other climate variables. The small range of the $\Delta$GPP
from the three ensemble members (Fig. 2) further indicates that the mismatch of Fdf among years does not essentially affect
the GPP estimation.

Compared with the *DF-PI-6H-CLIM* , *DF-PI-MON-CLIM* and *DF-PI-ENS* reconstructions, the *DF-PI-AERO* used atmo-
spheric radiative transfer model to partition the radiation into direct and diffuse components based on aerosol optical depth on
a 6-hourly time step during the entire period. In this way, the Fdf variation remains consistent with the variations of radiation
at all time scales. As expected, there is almost no bias in $\Delta$GPP at the pre-industrial period (Fig. 2, 4 and 5).

Considering the $\Delta$GPP bias could be also affected by the bias in total diffuse radiation due to the mismatch of Fdf and
total radiation, we further investigate the global mean diffuse radiation over all time steps in each reconstruction. We find that
different reconstructions have significant different mean diffuse radiation (Fig. 6). However, the difference in diffuse radiation
bias and the bias in $\Delta$GPP are not consistent (Fig. 2). For instance, the *DF-PI-6H-CLIM* and *DF-PI-ENS* reconstructions share
similar mean diffuse radiation but differ significantly in $\Delta$GPP . This difference implies that the mismatch between Fdf and
radiation is more important than the mean diffuse radiation over a long period.





### 4.3 Recommendations for defining a baseline pre-industrial climate forcing inclusive of diffuse light

As discussed above, different diffuse radiation reconstruction techniques can result in strongly different $\Delta$GPP in simulations. Therefore, it is important to have a reliable technique for scenario reconstruction and for diffuse radiation investigation.

By comparing the reconstructions used in this study, we argue that the *DF-PI-AERO* reconstruction, i.e. using atmospheric radiative transfer model to calculate the Fdf using aerosol and cloud information, can best simulate the GPP under no-aerosol scenario because the Fdf obtained from this method does not mismatch Fdf and solar radiation. However, to reconstruct the Fdf field in this way needs detailed aerosol and cloud information, which is not always available. In absence of such data, statistical methods are the alternative choice to do the reconstruction. In this case, our simulations have shown that the decrease of variability in Fdf due to any averaging processes can cause systematic mismatch between Fdf and incoming solar radiation,

which then biases the GPP. In contrast to the averaging methods, the mismatch in the *DF-PI-ENS* reconstruction is more random and the bias in *DF-PI-ENS* GPP is relatively small with small inter-simulation differences. Here we recommend that in future investigations of the impact of diffuse radiation in LSM offline simulations, the no-aerosol scenario Fdf should be calculated from aerosol and cloud information directly. When the information is not available, in ensemble simulations, repeating or randomly repeating the full Fdf time series from one or several pre-industrial years could become an acceptable alternative.

Despite that both reconstructions are acceptable in detecting diffuse radiation impacts, the impacts detected by the *DF-PI-AERO* and *DF-PI-ENS* reconstructions are not exactly the same. This is because that the *DF-PI-ENS* reconstruction implicitly eliminated Fdf changes caused by all factors including aerosols and clouds, while the *DF-PI-AERO* here has varying cloud information. Although we find similar $\Delta$GPP from the two reconstructions, we still cannot conclude with negligible cloud impacts because current cloud data remains very inaccurate.

### 265 5 Conclusions

For summary, in this study, we used different methods to reconstruct Fdf under no-anthropogenic-aerosol scenario and evaluated the influence of reconstruction methods on the diffuse radiation impacts on GPP using the ORCHIDEE_DF land surface model. We conclude that the traditional statistical methods by using a climatological average Fdf can cause 1-13 PgC yr$^{-1}$ bias on global GPP. To correctly simulate GPP, Fdf reconstructions need to retain its full variability. Based on our results, we

recommend to use pre-industrial aerosol information to calculate Fdf directly, or as an alternative in the absence of aerosol data, to use ensemble simulations driven by the original Fdf time series from pre-industrial years.

     Besides the experimental designs investigated in this study, it is also possible to use coupled simulations in ESMs to investigate the impacts of aerosols. In this way, the experiments can be better controlled and the climate-carbon feedback caused by the aerosol impacts can be investigated. However, due to the larger complexity of earth system models compared to LSMs, the

ESM experiments may suffer from larger uncertainties, which remain to get investigated.



# 6 Code and data availability

The code of the ORCHIDEE_DF used in this study is available at https://forge.ipsl.jussieu.fr/orchidee/wiki/GroupActivities/CodeAvalaibilityPublication/ORCHIDEE_DFv1.0_DFforc. The CRUJRA data and corresponding diffuse radiation fraction data used in this study is available at TRENDY ftp, details at http://sites.exeter.ac.uk/trendy/protocol

*Author contributions.* PC, OB and LL designed the project. YZ modified the model and ran all the simulations. NB provided the original diffuse radiation fraction field. YZ prepared the paper with contributions from all the co-authors.

*Competing interests.* The authors declare that they have no competing interests

*Acknowledgements.* The authors acknowledge support from European Research Council Synergy project SyG-2013-610028 IMBALANCE-P, H2020 EU CCICC project and the French ANR project China-Trend-Stream. The authors are grateful to the ORCHIDEE group for their kind help with the model and to the TRENDY group for providing the CRUJRA data.



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
