# Peer review of "How to reconstruct aerosol-induced diffuse radiation scenario for simulating GPP in land surface models? An evaluation of reconstruction methods with ORCHIDEE\_DFv1.0\_DFforc"

_Geoscientific Model Development, 2020_

## Referee Comment (RC1) · Anonymous Referee #1 · 12 Nov 2020

Yuan Zhang et al. "How to reconstruct diffuse radiation scenario for simulating GPP in land surface models?"

Recommendation to editor: Major

GENERAL: This paper compared different methods on reconducting spatiotemporal distribution of diffuse radiative fraction and explored the GPP responses to different diffuse conditions. Results show that the reconstruction of Fdf forcing fields need to be synchronous with aerosols and clouds amount. The topic is important and timely for exploring the diffuse fertilization effects. However, there are some important problems need to be solved on this paper:

(1) The whole study does not present any observations to validate the model and to

testify the importance of some Fdf modifications. The sensitivity of GPP to the changes of diffuse radiation should be validated against available observations. Mercado et al. (2009) provided a good example on how to perform such validations. Furthermore, several sensitivity experiments are performed with different settings of Fdf and show the consequent changes in GPP. However, such changes in GPP should be compared against observations to show which method can largely reduce modeling uncertainties.

(2) Figure 2a: the historical global mean Fdf are around 0.6-0.7 during 1900-2010, which is significantly vary from the results (0.4-0.5) from figure 3a by Mercado et al. (2009). What are the causes of such differences? Moreover, please show daily, seasonal and annual Fdf changes in supplement information so as to better validate predictions from different model. The global Fdf of DF-PI-ENS should be also added in Figure 1.

(3) Line 201: "This generally explains the spatial pattern of  $\triangle$ GPP detected in this study (Fig. 3a)." Fig.3a shows DF-PI-AERO underestimates significantly global GPP than DF-HIST, especially East Asia, Amazon and west Africa. Why do the different results appear? There are very small differences in Fdf between DF-HIST and DF-PI-AERO as shown in Figs 1a and 1d.

(4) Some of the conclusions are model dependent. For example, Lines 244-245, "This difference implies that the mismatch between Fdf and radiation is more important than the mean diffuse radiation over a long period." It remains unclear whether other models also support this conclusion. Again, the missing of observational validations makes this conclusion unconvincing.

SPECIFIC

Lines 38-45: Please add some recent references on aerosol-induced diffuse effects, such as Rap et al. (2018) and Yue and Unger (2018).

Lines 100-104: Are you using SW as input and calculate PAR for vegetation model?
Please explain how PAR and SW is connected in the model.

Figure 2: if possible, the interannual variations of Fdf from four reconstructions can be shown in Fig 2a.

Line 175: "dGPP" should be replaced as " $\Delta$ GPP".

Lines 207-209: "Because the solar zenith angle is large due to longer light path in atmosphere in the morning and afternoon, the Fdf is usually large in the morning and afternoon but low at midday (Iziomon and Aro, 1998)." These are conflicting with lines 124-125, which say: "This method accounts for the periodical diurnal increase of Fdf from morning to mid-day and its decrease from mid-day to afternoon."

Reference: Mercado, L.M., Bellouin, N., Sitch, S., Boucher, O., Huntingford, C., Wild, M., Cox, P.M., 2009. Impact of changes in diffuse radiation on the global land carbon sink. Nature 458, 1014-U1087. Rap, A., Scott, C.E., Reddington, C.L., Mercado, L., Ellis, R.J., Garraway, S., Evans, M.J., Beerling, D.J., MacKenzie, A.R., Hewitt, C.N., Spracklen, D.V., 2018. Enhanced global primary production by biogenic aerosol via diffuse radiation fertilization. Nature Geoscience 11, 640-+. Yue, X., Unger, N., 2018. Fire air pollution reduces global terrestrial productivity. Nature Communications 9.

---

## Short Comment (SC1) · 14 Nov 2020

Dear authors,

in my role as Executive editor of GMD, I would like to bring to your attention our Editorial version 1.2:

https://www.geosci-model-dev.net/12/2215/2019/

This highlights some requirements of papers published in GMD, which is also available on the GMD website in the 'Manuscript Types' section:

http://www.geoscientific-model-development.net/submission/manuscript_types.html

In particular, please note that for your paper, the following requirement has not been

met in the Discussions paper:

- "The main paper must give the model name and version number (or other unique identifier) in the title."

- "If the model development relates to a single model then the model name and the version number must be included in the title of the paper. If the main intention of an article is to make a general (i.e. model independent) statement about the usefulness of a new development, but the usefulness is shown with the help of one specific model, the model name and version number must be stated in the title. The title could have a form such as, "Title outlining amazing generic advance: a case study with Model XXX (version Y)"."

As your analysis is performed with one model only, add the name and version number of the model to the title of your publication in you revised version.

Yours,

Astrid Kerkweg

---

## Referee Comment (RC2) · Anonymous Referee #2 · 30 Nov 2020

This manuscript presents a study on how the errors in the reconstructed fraction of diffuse radiaiton (Fdf) would affect global GPP estimation using the ORCHIDEE_DF land surface model. The authors have investigated a few methods to reconstruct Fdf under pre-industrial aerosol emission conditions and shown that different reconstruction method may result in diverse Fdf and large biases in global GPP estimation. The study may be useful for the land surface modeling and global carbon cycling research community and thus worth publishing, however I have a few concerns that I feel have to be addressed.

First, the title is not accurate. Since clouds can be a major contributor to diffuse radiation, the paper is actually about aerosol-induced diffuse radiation scenario.

Second, the presentation quality of this paper needs to be improved. A lot of important

details are missing. For example, L49: what are the time spans of 'historical' and 'pre-industrial'? L106-112 is an extremely long sentence. Consider breaking it into shorter ones. The calculation of Fdf is confusing. How did you make the 'atmospheric radiative transfer calculations'? How do you make sure Fdf is consistent with the CRUJRA data, while using aerosol data from other sources? For which years the reconstructing methods were applied? Which years were used for ORCHIDEE_DF runs? These all need to be stated in the methods section.

Third, which I think the most problematic, the reconstructing methods don't remove the huge cloud impacts on Fdf, thus implicitly apply the cloud conditions in the base years in 1901-1920 to other years. Therefore, if I understand it correctly, the work doesn't actually study the aerosol-induce changes of Fdf. In addition, as stated in L137, 'Except the Fdf field, all these simulations use the same climate and land use maps which vary throughout the simulations'. Usually the downward shortwave radiation covary with Fdf; in other words, if Fdf is changed ('reconstructed'), the total downward radiation should also be changed accordingly — this is why a lot of empirical method can successfully estimate diffuse radiation from the total downward radiation with promising accuracy (e.g., see Berrizbeitia, S.E.; Jadraque Gago, E.; Muneer, T. Empirical Models for the Estimation of Solar Sky-Diffuse Radiation. A Review and Experimental Analysis. ÂăEnergiesÂă2020, Âă13, 701.). Actually these empirical methods are efficient options for estimate (or reconstruct) Fdf with historical climate fields, although they are not able to distinguish the contribution of anthropogenic aerosols.

Overall, the authors needs to further justify their methods, otherwise the story of impacts on GPP really lacks a solid foundation.

Overall, the authors needs to further justify their methods, otherwise the story of impacts on GPP really lacks a solid foundation.
* * *

---

## Author Comment (AC1) · 11 Jan 2021

**Response to Reviewer #1**

**General comment:**

*"This paper compared different methods on reconstructing spatiotemporal distribution of diffuse radiative fraction and explored the GPP responses to different diffuse conditions. Results show that the reconstruction of Fdf forcing fields need to be synchronous with aerosols and clouds amount. The topic is important and timely for exploring the diffuse fertilization effects. However, there are some important problems need to be solved on this paper:"*

[Response] We thank the Reviewer for the review, comments and suggestions, which helped us to improve our manuscript. We have addressed all the suggestions and comments in our revision. Please find below the Reviewer's comments (italics), followed by our responses (roman font), with red color indicating relevant changes in the manuscript. We hope that the revised manuscript addresses all the issues raised by the Reviewer.

**Comments:**

*"(1) The whole study does not present any observations to validate the model and to testify the importance of some Fdf modifications. The sensitivity of GPP to the changes of diffuse radiation should be validated against available observations. Mercado et al. (2009) provided a good example on how to perform such validations. Furthermore, several sensitivity experiments are performed with different settings of Fdf and show the consequent changes in GPP. However, such changes in GPP should be compared against observations to show which method can largely reduce modeling uncertainties."*

[Response] We thank the Reviewer for this comment. Indeed, all models need to be validated or at least evaluated against observations before use. In this manuscript, we did not put the details of the validation. However, in our previous study (Zhang et al. 2020), we have extensively validated our model using observations from 159 flux sites and shown that ORCHIDEE_DF is able to reproduce the diffuse radiation fertilization effect in most vegetation types (Figure R1). We have mentioned this in the manuscript

(Line 86-88). Therefore, it is reasonable to use the current model to investigate the diffuse radiation impact.

"*(2) Figure 2a: the historical global mean Fdf are around 0.6-0.7 during 1900-2010, which is significantly vary from the results (0.4-0.5) from figure 3a by Mercado et al. (2009). What are the causes of such differences? Moreover, please show daily, seasonal and annual Fdf changes in supplement information so as to better validate predictions from different model. The global Fdf of DF-PI-ENS should be also added in Figure 1.*"

[Response] Thanks for this question. We also noted this difference on Fdf between this study and Mercado et al. (2009). In this study, the mean Fdf is calculated by averaging all the daytime 6-hourly Fdf over the land regions in each year. Mercado et al. (2009) did not explain how the average of Fdf was calculated, but they have done an adjustment of radiation (reducing the diffuse radiation under cloudy conditions) in their study, as described in their methods section. This might cause the difference in mean Fdf between the two studies. In this study we used the radiation from the CRUJRA dataset directly as this dataset is observationally-based and has widely been used in global simulations.

To address this suggestion, we have shown the diurnal and seasonal cycles of Fdf of some selected grid points in the supplementary (Fig. R2).

The DF-PI-ENS has three simulations, one of them, DF-PI-1901 has exactly the same Fdf pattern as Figure 1a, and DF-PI-1905, DF-PI-1916 has similar spatial variability as DF-PI-1901, which has been added to the supplementary (Fig. R3).

*(3) Line 201: "This generally explains the spatial pattern of ∆GPP detected in this study (Fig. 3a)." Fig.3a shows DF-PI-AERO underestimates significantly global GPP than DF-HIST, especially East Asia, Amazon and west Africa. Why do the different results appear? There are very small differences in Fdf between DF-HIST and DF-PIAERO as shown in Figs 1a and 1d.*

**[Response]** Thanks for this question, Fig. 1 and Fig. 3 correspond to different time scales and periods. Figure 3 is the mean aerosols impact on GPP during 1961-2010, when there are significant anthropogenic aerosol emissions in the regions mentioned by the Reviewer. If we compare the mean difference in Fdf during this period, *DF-PI-AERO* should have lower Fdf. In contrast, Figure 1 shows the different variability of Fdf in different reconstructions. Because the long-term mean Fdf of the reconstructions should always be the same due to the average method we used, we showed only the first time step in Figure 1, which is in 1901. We chose this time because the aerosol levels are similar between *DF-HIST* and the other reconstructions so the *DF-HIST* Fdf can be a reference to check the variability of Fdf in different reconstructions.

*(4) Some of the conclusions are model dependent. For example, Lines 244-245, "This difference implies that the mismatch between Fdf and radiation is more important than the mean diffuse radiation over a long period." It remains unclear whether other models also support this conclusion. Again, the missing of observational validations makes this conclusion unconvincing.*

**[Response]** Thanks for this point. Indeed, the results could be model dependent. However, as discussed in Section 4.2, if the model represents well the mechanism how Fdf affects photosynthesis, it should get biased GPP when using mismatched Fdf and SWdown in a simulation. Nevertheless, we added in Line 245: "Nevertheless, GPP always differs between LSMs. The magnitude of the GPP bias due to the mismatch between Fdf and SWdown detected here is only for ORCHIDEE_DF model and needs to be further investigated in other LSMs. Nevertheless, the framework that we propose is applicable to any LSM" For the validation, please check the response to Comment 1.

**Specific comments:**

*Lines 38-45: Please add some recent references on aerosol-induced diffuse effects, such as Rap et al. (2018) and Yue and Unger (2018).*

**[Response]** Thanks, the references have been added in Line 42: "Rap et al. (2018) and Yue and Unger (2018) also used simulation under different scenarios, but with different reconstruction of diffuse radiation." And Line 251: "Similar methods have been used in Rap et al. (2018) and Yue and Unger (2018)."

*Lines 100-104: Are you using SW as input and calculate PAR for vegetation model? Please explain how PAR and SW is connected in the model.*

**[Response]** Yes, SWdown is used as input. A factor of 0.5 is used to calculate PAR from SW.

*Figure 2: if possible, the interannual variations of Fdf from four reconstructions can be shown in Fig 2a.*

**[Response]** Thanks, we have added the reconstruction Fdf in Figure 2a (Figure R4). The description of the updated figure has been added to the manuscript (Line 147): "Because the no-anthropogenic-aerosol reconstructions DF-PI-6H-CLIM, DF-PI-ENS, DF-PI-MON-CLIM use the volcano-free years during 1901-1920, they produce the same or very similar global yearly mean Fdf around 0.615 during the entire study period. For the DF-PI-AERO reconstruction, the Fdf increased by about 0.005 after the 1950s, which is not comparable to the increase of Fdf in DF-HIST. This increase is mainly due to the changes in cloudiness and natural aerosols."

*Line 175: "dGPP" should be replaced as "ΔGPP".*

**[Response]** It has been corrected accordingly.

*Lines 207-209: "Because the solar zenith angle is large due to longer light path in*

*atmosphere in the morning and afternoon, the Fdf is usually large in the morning and afternoon but low at midday (Iziomon and Aro , 1998)." These are conflicting with lines 124-125, which say: "This method accounts for the periodical diurnal increase of Fdf from morning to mid-day and its decrease from mid-day to afternoon."*

**[Response]** Thanks for pointing out this error in Lines 124-125, which is now corrected: "This method accounts for the periodical diurnal decrease of Fdf from morning to mid-day and its increase from mid-day to afternoon."

**References**

Zhang, Y., Bastos, A., Maignan, F., Goll, D., Boucher, O., Li, L., Cescatti, A., Vuichard, N., Chen, X., Ammann, C., Arain, M. A., Black, T. A., Chojnicki, B., Kato, T., Mammarella, I., Montagnani, L., Roupsard, O., Sanz, M. J., Siebicke, L., Urbaniak, M., Vaccari, F. P., Wohlfahrt, G., Woodgate, W., and Ciais, P.: Modeling the impacts of diffuse light fraction on photosynthesis in ORCHIDEE (v5453) land surface model, Geosci. Model Dev., 13, 5401–5423, https://doi.org/10.5194/gmd-13-5401-2020, 2020.

[Figure]

**Figure R1: Observed GPP and GPP modeled by ORCHIDEE trunk and ORCHIDEE_DF under cloudy (diffuse light fraction >0.8) and sunny (diffuse light fraction<0.4) conditions at selected sites (with relatively long time series) from each PFT. Figure from Zhang et al. (2020)**

[Figure]

Figure R2. The diurnal cycle of Fdf on 1901-01-01 (a) and (b), and the seasonal cycle of Fdf in 1901 (c) and (d) in different reconstructions at selected grids. The open markers in (a) and (b) are night time values filled with 1.

[Figure]

Figure R3. Same as Fig 1 but for DF-PI-1901, DF-PI-1905 and DF-PI-1916

[Figure]

Figure R4. Time series of (a) global mean Fdf of different reconstructions and (b) ΔGPP between DF-HIST and no-anthropogenic-aerosol scenarios. The shaded area along the red curve in (b) indicates the range of the three ensemble members of the DF-PI-ENS simulations. The DF-PI-MON-CLIM has the same mean Fdf as DF-PI-6H-CLIM, thus not shown in (a). This is the update of Fig 2 to include the mean Fdf of PI level reconstructions.

---

## Author Comment (AC2) · 11 Jan 2021

**Response to Reviewer #2**

**General comment:**

*"This manuscript presents a study on how the errors in the reconstructed fraction of diffuse radiaiton (Fdf) would affect global GPP estimation using the ORCHIDEE_DF land surface model. The authors have investigated a few methods to reconstruct Fdf under pre-industrial aerosol emission conditions and shown that different reconstruction method may result in diverse Fdf and large biases in global GPP estimation. The study may be useful for the land surface modeling and global carbon cycling research community and thus worth publishing, however I have a few concerns that I feel have to be addressed."*

[Response] We thank the Reviewer for the review, comments and suggestions, which helped us to improve our manuscript significantly. We have addressed all the suggestions and comments in our revision. Please find below the Reviewer's comments (italics), followed by our responses (roman font), with red color indicating relevant changes in the manuscript. We hope that the revised version addresses all the issues raised by the Reviewer.

**Comments:**

*First, the title is not accurate. Since clouds can be a major contributor to diffuse radiation, the paper is actually about aerosol-induced diffuse radiation scenario.*

[Response] Thanks for this suggestion, we have changed the title to "How to reconstruct aerosol-induced diffuse radiation scenario for simulating GPP in land surface models? An evaluation of reconstruction methods with ORCHIDEE_DFv1.0_DFforc"

*"Second, the presentation quality of this paper needs to be improved. A lot of important details are missing. For example, L49: what are the time spans of 'historical' and 'preindustrial'? L106-112 is an extremely long sentence. Consider breaking it into shorter ones. The calculation of Fdf is confusing. How did you make the 'atmospheric*

*radiative transfer calculations'? How do you make sure Fdf is consistent with the CRUJRA data, while using aerosol data from other sources? For which years the reconstructing methods were applied? Which years were used for ORCHIDEE_DF runs? These all need to be stated in the methods section."*

**[Response]** Thanks for the suggestions. We have rephrased these sentences and added more information in the manuscript (Line 49): "e.g., the scenario with actual anthropogenic aerosol emissions and the one with aerosol emissions at pre-industrial level (before or in early 20$^{th}$ century)". (Lines 106-112): "For the sake of investigating the effect of diffuse radiation with a framework consistent with the TRENDY simulation protocol (http://sites.exeter.ac.uk/trendy/), a new Fdf field during 1900-2017 was calculated along with the above-mentioned climate variables at the same spatial and temporal resolutions. The radiative transfer calculations to obtain the Fdf field are based on monthly-averaged distributions of tropospheric and stratospheric aerosol optical depth, and 6-hourly distributions of cloud fraction. The tropospheric aerosol optical depth of each aerosol type is from the HadGEM2-ES historical and RCP8.5 simulations (Bellouin et al. 2011). To correct the biases in HadGEM2-ES, tropospheric aerosol optical depths are scaled over the entire period to match the global and monthly averages obtained by the CAMS Reanalysis of atmospheric composition for the period 2003-2017 (Inness et al. 2019), which assimilates satellite retrievals of aerosol optical depth. The stratospheric aerosol optical depth is from the climatology by Sato et al. (1993), which has been updated to 2012. Years after 2012 are assumed to be background years without significant influence of volcanoes and the stratospheric aerosol optical depth is assumed to be the same as a recent background year 2010. This assumption is supported by the Global Space-based Stratospheric Aerosol Climatology time series (1979-2016; Thomason et al. 2018). The time series of cloud fraction is obtained by scaling the 6-hourly cloud distributions simulated by the Japanese Reanalysis (JRA; Kobayashi et al. 2015) to match the monthly-averaged cloud cover in the CRU TS v4.03 dataset (Harris et al. 2014). Surface radiative fluxes calculation accounts for aerosol-radiation interactions from both tropospheric and stratospheric

aerosols, and for aerosol-cloud interactions from tropospheric aerosols, except mineral dust. The radiative effects of aerosol-cloud interactions are assumed to scale with the radiative effects of aerosol-radiation interactions, and regional scaling factors derived from HadGEM2-ES are used in the calculation. Atmospheric constituent other than aerosols and clouds are set to a constant standard mid-latitude summer atmosphere, but their variations do not affect the diffuse fraction of surface shortwave fluxes." (Line 115): "Based on the assumption that this sample is representative of the pre-industrial aerosol conditions, four methods are used to reconstruct 0.5 ° x 0.5 ° 6-hourly pre-industrial Fdf field and corresponding simulations are set up for the period 1901-2017."

*Third, which I think the most problematic, the reconstructing methods don't remove the huge cloud impacts on Fdf, thus implicitly apply the cloud conditions in the base years in 1901-1920 to other years. Therefore, if I understand it correctly, the work doesn't actually study the aerosol-induce changes of Fdf. In addition, as stated in L137, 'Except the Fdf field, all these simulations use the same climate and land use maps which vary throughout the simulations'. Usually the downward shortwave radiation covary with Fdf; in other words, if Fdf is changed ('reconstructed'), the total downward radiation should also be changed accordingly ă˘A˘T this is why a lot of empirical method can successfully estimate diffuse radiation from the total downward radiation with promising accuracy (e.g., see Berrizbeitia, S.E.; Jadraque Gago, E.; Muneer, T. Empirical Models for the Estimation of Solar Sky-Diffuse Radiation. A Review and Experimental Analysis. ¢aEnergies¢a2020,¢a13, 701.). Actually these empirical methods are efficient options for estimate (or reconstruct) Fdf with historical climate fields, although they are not able to distinguish the contribution of anthropogenic aerosols.*

**[Response]** Thanks very much for this comment.

First, the full impacts of aerosol and clouds should include the impacts from changes in both light quantity (SWdown) and light quality (Fdf). In this study, we focus on the impacts of light quality only, which is similar to the study of Mercado et al. (2009).

That is why we kept the SWdown unchanged in all the simulations. To understand the full impacts of aerosol-induced radiation changes, SWdown changes need to also get considered. However, it is out of the scope of this study. We have added a statement of this point in the manuscript (Lines 135) "It should be noted that all the simulations in this study use the same SWdown field because the target of this study is to understand the impact from aerosol-induced radiation quality, i.e. Fdf, changes only. In reality, the aerosols and clouds also cause a coincident change in radiation quantity, i.e. SWdown, which is important to consider when investigating the full impacts for aerosols. But it is out of the scope of this study."

Second, we totally agree with the reviewer that clouds can strongly affect Fdf and should be considered when doing the reconstruction. However, current statistical reconstruction methods are not able to decompose the impacts of aerosols from clouds, therefore, the impacts of clouds on Fdf are omitted in the DF-PI-6H-CLIM, DF-PI-MON-CLIM and DF-PI-ENS reconstructions. In contrast to the statistical method-based reconstructions, the variation of cloud is considered the DF-PI-AERO reconstruction because it used varying clouds in the Fdf calculation.

Although the impacts of clouds are not included in the statistical reconstructions, we still can compare the GPP among the simulations in the early 1900s to evaluate the reconstruction methods because the cloud fractions are at the same level for the scenarios with and without anthropogenic aerosol emissions. It is clear that the disagreement among simulations start from 1901 (Fig. 2b). Furthermore, the DF-PI-ENS and DF-PI-AERO reconstructions have different cloud impacts after the 1920s but show similar global GPP (Fig 2b). Therefore, the difference in the consideration of clouds should not be the reason of the GPP bias detected. The conclusion that the smooth of Fdf in the reconstruction caused the GPP bias remains valid. Nevertheless, considering the changes of cloudiness is still important for an accurate investigation of the impact of Fdf changes. The impact of clouds is discussed in Lines 260-264: "Despite that both reconstructions are acceptable in detecting diffuse radiation impacts, the impacts detected by the DF-PIAERO and DF-PI-ENS reconstructions are not exactly the same. This is because that the DF-PI-ENS reconstruction implicitly eliminated Fdf

changes caused by all factors including aerosols and clouds, while the DF-PI-AERO here has varying cloud information. In this study, the impacts of cloud difference on GPP are much smaller than the bias caused by the problematic Fdf reconstructions (Fig. 2). However, we still cannot conclude with negligible cloud impacts because current cloud data remains very inaccurate."

---

## Author Comment (AC3) · 11 Jan 2021

Thanks for the comment, we have changed the title of the manuscript to include the model name:

"How to reconstruct aerosol-induced diffuse radiation scenario for simulating GPP in land surface models?    An evaluation of reconstruction methods with OR-CHIDEE_DFv1.0_DFforc"